# Incidence of depression and anxiety in children and adolescents following COVID-19 infection

Jaewhan Kim[1]*, Carson R. Ewing[2], Ashlee Larson[2], Chathuri Illapperuma-Wood[2], Aaron Fischer[2], Emeka Elvis Duru[3], Youngwoo Kim[2], Fernando Wilson[4]

1 Department of Physical Therapy, University of Utah, Salt Lake City, Utah, United States of America, 2 Department of Educational Psychology, University of Utah, Salt Lake City, Utah, United States of America, 3 Department of Pharmacotherapy, University of Utah, Salt Lake City, Utah, United States of America, 4 Department of Population Health Sciences, University of Utah, Salt Lake City, Utah, United States of America

* Jaewhan.kim@utah.edu

## Abstract

While the COVID-19 pandemic has significantly impacted mental health, limited information exists on the relationship between COVID-19 infection and mental health disorders in children and adolescents. This study explored the association between COVID-19 infection, infection severity, and the risk of depression and anxiety among school-aged youth. Data from the 2019–2021 Utah All Payers Claims Database (APCD) was used to identify children and adolescents (aged 6–15 years in 2019) covered by private insurance or Medicaid. Depression and anxiety diagnoses in 2021 were identified using diagnosis codes. Entropy balancing was applied to balance baseline differences between youth with and without a COVID-19 infection in 2020. Weighted logistic regression was conducted to identify factors associated with incidence of depression and anxiety. A total of 154,579 subjects were included, with an average age of 10.8 years (SD: 2.8); 48% were female. Youth with a COVID-19 infection had a 49% higher likelihood of having new depression or anxiety diagnoses in 2021 (OR = 1.49, p < 0.01). Compared to youth without COVID-19 infection, those with moderate symptoms had a 40% higher likelihood (OR = 1.40, p < 0.01), and those with severe symptoms had a 60% higher likelihood (OR = 1.60, p < 0.01) of developing depression or anxiety. This population-level study found that COVID-19 infection, especially severe cases, is associated with an increased risk of depression and anxiety in children and adolescents. These findings underscore the importance of targeted mental health interventions for youth impacted by the infection.

## Introduction

The prevalence of mental health disorders among youth has surged during the COVID-19 pandemic, with significant increases in depression and anxiety. The proportion of youth experiencing a major depressive episode rose from 9.1% in 2019 to

**Data availability statement:** Due to legal and ethical restrictions imposed by the Utah Department of Health and Human Services (third-party organization), the University of Utah's Institutional Review Board, and the Utah Resource for Genetic and Epidemiologic Research (RGE), we are unable to publicly share even de-identified individual-level data used in this study. Specifically, these data contain potentially identifying and sensitive patient health information protected under state regulations and institutional policies. Data access is managed by the RGE, and requests for data can be directed to the University of Utah Institutional Review Board/ Data Access Committee via the RGE website (https://rge.utah.edu) or through direct contact at: Institutional Review Board (IRB) Office University of Utah 75 South 2000 East, Salt Lake City, Utah 84112 Phone: 801-581-3655 Email: irb@hsc.utah.edu Utah Resource for Genetic and Epidemiologic Research (RGE) University of Utah Research Administration Building, 75 S 2000 E room 207, Salt Lake City, UT 84132 Phone: 801-581-6351 Email: rge@hsc.utah.edu.

**Funding:** The author(s) received no specific funding for this work.

**Competing interests:** The authors have declared that no competing interests exist.

18.33% in 2023, as reported by Mental Health America [1,2]. The rise in mental illness among children (aged 6–11) and adolescents (aged 12–17) during this period is likely a result of several interconnected factors. Key contributors include the pandemic's direct impact on daily life, such as school closures, isolation from peers, and disruptions to family routines. These challenges have likely had long-term consequences for the mental health and well-being of young people [3,4]. Moreover, COVID-19 infection itself may play a significant role in the development of mental health disorders among children and adolescents. Although the relationship between COVID-19 infection and mental health outcomes in youth remains inconclusive, prior research has shown varying results. Some studies suggest that contracting COVID-19 may elevate the risk of mental health disorders, including anxiety and depression, among youth [5,6]. Research has shown that children and adolescents who contracted COVID-19 may have been more likely to experience heightened mental health challenges [7]. For instance, one study found that children with pre-existing mental health conditions experienced exacerbated symptoms following COVID-19 infection [8].

However, other studies have reported no significant relationship between COVID-19 infection and mental health outcomes in children [9,10], with one study suggesting the need for a holistic approach to adolescent mental health, asserting that there is no clear evidence linking COVID-19 infection to an increased risk of mental health problems [10].

Despite the potential impact of COVID-19 on youth mental health, a notable gap exists in the literature exploring the relationship between COVID-19 infection and subsequent mental health outcomes in children and adolescents. Previous studies have often involved small sample sizes, limiting the generalizability of their findings, and many have overlooked the severity of COVID-19 infection as a potential factor influencing the development of mental health disorders in this population [7].

Children and adolescents are at a critical stage of emotional, social, and cognitive development, and disruptions during this period can have profound and long-lasting effects on academic performance, mental health, and overall well-being [11]. Understanding the role that COVID-19 infection might play in exacerbating mental health issues during this developmental phase is crucial for developing targeted interventions and support mechanisms for affected youth.

Given these gaps, this study aims to examine the potential causal relationship between COVID-19 infection, its severity, and the associated risks of depression and anxiety among school-aged children. By utilizing a state-level population database covering the period from 2019 to 2021, this research seeks to provide valuable insights into how COVID-19 infection may contribute to the mental health challenges faced by children and adolescents, offering a more comprehensive understanding of the pandemic's long-term impacts on youth mental health.

## Materials and methods

### Data

The study was conducted in accordance with the Declaration of Helsinki, and approved by the Institutional Review Board (or Ethics Committee) of University of

Utah (protocol code IRB_00151091 and November 09, 2023). This study was a retrospective cohort analysis utilizing data from the 2019–2021 Utah All Payers Claims Database (APCD) to investigate the causal relationship between COVID-19 infection and the incidence of depression and anxiety among school-aged children and adolescents. The APCD encompasses over 70% of Utah's population with private insurance, Medicaid, and Medicare Advantage. The APCD consists of two key files: the enrollment file and the claims file. The enrollment file includes information on insurance coverage start and end dates, birth dates, gender, race/ethnicity, and the type of insurance (private, Medicaid, or Medicare). The claims file provides data on service/visit dates, claim types (medical, pharmacy, and dental), place of service, bill types, provider specialty, drug names, national drug codes (NDC), and healthcare costs, including reimbursed amounts and co-payments. Additionally, it includes diagnostic and procedural codes following the Current Procedural Terminology (CPT) and International Classification of Diseases, 10th Revision (ICD-10) standards. More information about the APCD can be found elsewhere [12–14]. Data used in this study were de-identified, which led to an exemption determination from the Institutional Review Board (IRB) at the University of Utah (IRB 00151091). The data were accessed between December 3, 2024, and February 15, 2025, for this research. The authors did not have access to identifiable information of the subjects during or after the data access period.

## Subjects

The study population consisted of school-aged children and adolescents in Utah, aged 6–15 years in 2019, with continuous health insurance enrollment from 2019 through 2021. Children and adolescents who were 15 years old in 2019 were 17 years old in 2021 and remained in the school-age category. Children and adolescents who were eligible for Medicare due to disability (i.e., dual eligible) were excluded from this analysis, as the pediatric Medicare population is extremely small and typically consists of individuals with rare and complex clinical conditions that are not representative of the general child and adolescent population.

## Outcome

The primary dependent variable assessed was the occurrence of depression and anxiety in 2021, assessed as a binary variable (Yes/No). Depression and anxiety were identified using ICD-10 codes according to the Centers for Medicare and Medicaid Services (CMS) [15], and Agency for Healthcare Research and Quality (AHRQ) Clinical Classifications Software Refined (CCSR) guidelines [16]. Depression was identified using ICD-10 codes (F06.31-F06.34, F32-F33.9), explicitly excluding F34.1 (dysthymia) due to its diagnostic requirement of prolonged symptom duration which exceeds the one-year follow-up of our study. Anxiety was identified using ICD-10 codes (F06.4, F12.180-F16.980, F18.180-F19.980, F40.00-F41.9, F93.0, F94.0). Alcohol-related disorders (ICD-10 codes F10.xx) were excluded due to their limited clinical relevance within the studied pediatric population.

## Independent variable

Two independent variables were created. The first variable was binary, indicating COVID-19 infection (Yes/No) in 2020, while the second variable had three categories reflecting the severity of COVID-19 infection: no infection, mild infection, and severe infection. COVID-19 infection was identified using a combination of ICD-10 codes (J1281, J1282, U071, U099, B948, B9729, Z8616, O985, Z20828) and CPT codes (86413, 86328, 86769, 87426, 87428, 87635, 87636, 87637, 87811, 87913, C9803) [15–17]. Severity of infection was determined by healthcare utilization in conjunction with COVID-19 diagnosis. Mild infections were classified as COVID-19 diagnoses without any emergency room (ER) visits or hospitalizations, while severe infections were defined as those associated with an ER visit or hospital admission within seven days before or after the COVID-19 diagnosis [15].

## Covariates

A total of 27 chronic conditions were considered, and 15 covariates were selected using the Least Absolute Shrinkage and Selection Operator (LASSO), a machine learning approach. These covariates included age category (6–10 vs. 11–15), gender (male vs. female), race/ethnicity (non-Hispanic White, Hispanic, non-Hispanic Black, non-Hispanic Asian/Pacific Islander/American Indian, Unknown), insurance type (private insurance vs. Medicaid), sleep disorder (Yes/No), Attention-Deficit/Hyperactivity Disorder (ADHD) (Yes/No), pain diagnosis (Yes/No), blindness (Yes/No), any cancer diagnosis (Yes/No), asthma (Yes/No), chronic headaches including migraines (Yes/No), allergy reactions including anaphylaxis (Yes/No), hearing loss (Yes/No), neurocognitive disorder (Yes/No), and obesity (Yes/No).

## Statistical approach

Covariate selection was conducted using LASSO, resulting in 15 variables being selected from the 27 chronic conditions considered. The selected lambda (λ) based on 10-fold cross-validation was 0.0003, with a deviance of 0.446.

All covariates, except for blindness, showed statistically significant differences between subjects with and without COVID-19 infection. To account for these differences, which could impact the incidence of depression and anxiety in 2021, Entropy Balancing (EB) was employed to equalize these baseline characteristics [15,18]. A common approach for balancing covariates is Propensity Score Matching (PSM), which selects matched cases and controls. In contrast, EB utilizes all subjects available in the study to create weights for balancing. A study reported that EB is more effective at balancing covariates than PSM and is doubly robust [19]. Before and after EB, standardized mean differences (SMD) of the covariates were compared, with all SMD values after EB being less than 0.1, indicating good balance and effectively eliminating observed confounders that could affect the outcome (S1 Appendix).

Weighted summary statistics, including mean, standard deviation (SD), and percentages, were used to summarize the characteristics of the study subjects. T-tests were conducted for continuous variables, and Chi-square tests were used for categorical variables to assess statistical differences in the summary table. Weighted logistic regression was employed to identify factors associated with the incidence of depression and anxiety. The first logistic regression included subjects with and without COVID-19 infection as the independent variable, while the second logistic regression categorized subjects based on their COVID-19 symptoms: no infection, moderate symptoms, and severe symptoms. A p-value of less than 0.05 was considered statistically significant. Stata 18.5 was used for the analysis.

## Results

A total of 894,884 children aged 0–17 years were identified in 2019. Among them, 352,087 had 36 months of continuous enrollment, and 217,434 children aged 6–15 years were selected for the study. A total of 33,159 subjects with depression or anxiety diagnoses in 2019 and 2020 were excluded. Additionally, 29,696 subjects with a COVID-19 diagnosis in 2021 were also excluded from the analysis. The final sample size for the analysis was 154,565 subjects.

After entropy weighting, the average age in 2019 was 10.7 years (SD = 2.8), with approximately 56% of subjects in the 11–15 age group. Females comprised 48% of the sample, and about 26% were covered by Medicaid. Among the subjects, 2.5% had a sleep disorder, 5% had an ADHD diagnosis, and 7.4% had an asthma diagnosis in 2019 (Table 1).

## Discussion

This study contributes to the small body of literature that examines mental health detriments of youth infected by COVID-19 infection. It demonstrates a relationship between COVID-19 infections among youth and elevated rates of depression and anxiety. Overall, youth with a history of COVID-19 infection exhibited 49% higher rates of mental health concerns compared to youth without infection. Children and adolescents who experienced a severe COVID-19 infection experienced 59% higher rates of depression and anxiety. There is limited information about impact of COVID-19 infection on

**Table 1. Characteristics of subjects with and without COVID-19 infection.**

| | COVID-19 infection | | | |
| --- | --- | --- | --- | --- |
| | No | Yes | Total | p-value |
| N | 119,424(77.3%) | 35,141 (22.7%) | 154,565(100.0%) | |
| Age in 2019 (continuous) | 10.707 (2.805) | 10.843 (2.852) | 10.738(2.817) | <0.001 |
| Age category | | | | 0.781 |
| 6-10 | 52,657 (44.1%) | 15,464 (44.0%) | 68,121(44.1%) | |
| 11-15 | 66,767 (55.9%) | 19,677 (56.0%) | 86,444(55.9%) | |
| Female | 57,663 (48.3%) | 16,952 (48.2%) | 74,615 (48.3%) | 0.888 |
| Medicaid | 31,309 (26.2%) | 9,019 (25.7%) | 40,328 (26.1%) | 0.0.051 |
| Race/Ethnicity | | | | 0.309 |
| Non-Hispanic White | 16,734 (14.0%) | 4,785 (13.6%) | 21,519 (13.9%) | |
| Hispanic | 9,787 (8.2%) | 2,823 (8.0%) | 12,610 (8.2%) | |
| Non-Hispanic Black | 730 (0.6%) | 215 (0.6%) | 945 (0.6%) | |
| Non-Hispanic others | 2,079 (1.7%) | 617 (1.8%) | 2,696 (1.7%) | |
| Unknown | 90,094 (75.4%) | 26,701 (76.0%) | 116,795 (75.6%) | |
| Sleep disorder | 2,962 (2.5%) | 781 (2.5%) | 3,743 (2.5%) | 0.993 |
| ADHD | 6,031 (5.0%) | 1,590 (5.1%) | 7,621 (5.0%) | 0.994 |
| Pain | 25,313 (21.2%) | 6,676 (21.2%) | 31,989 (21.2%) | 0.969 |
| Blindness | 2,978 (2.5%) | 785 (2.5%) | 3,763 (2.5%) | 0.999 |
| Cancer | 8,839 (7.4%) | 2,331 (7.4%) | 11,170 (7.4%) | 0.986 |
| Asthma | 8,850 (7.4%) | 2,334 (7.4%) | 11,184 (7.4%) | 0.985 |
| Headache | 5,343 (4.5%) | 1,409 (4.5%) | 6,752 (4.5%) | 0.990 |
| Severe allergy | 12,310 (10.3%) | 3,246 (10.3%) | 15,556 (10.3%) | 0.987 |
| Hearing loss | 1,691 (1.4%) | 446 (1.4%) | 2,137 (1.4%) | 0.993 |
| Neurocognitive disorders | 9,593 (8.0%) | 2,529 (8.0%) | 12,122 (8.0%) | 0.994 |
| Obesity | 8,067 (6.8%) | 2,205 (6.8%) | 10,272 (6.8%) | 0.991 |

Among the total subjects, 22.7% had COVID-19 infection in 2020, and 9.2% of those with COVID-19 experienced new depression or anxiety in 2021. In contrast, 5.3% of subjects without COVID-19 had new incidences of depression or anxiety in 2021. Of all subjects in 2020, 12.3% had moderate symptoms, while 10.4% had severe symptoms. Among those with moderate symptoms, 8.3% experienced new depression or anxiety, compared to 10.2% of those with severe symptoms (p<0.01) (Table 2).

**Table 2. Incidence of depression and anxiety in 2021 by COVID-19 infection status.**

| | COVID-19 infection in 2020 | | | |
| --- | --- | --- | --- | --- |
| | No | Yes | Total | p-value |
| N | 119,424 (77.3%) | 35,141 (22.7%) | 154,565 (100.0%) | |
| Incidence of depression or/and anxiety in 2021 | 6,276 (5.3%) | 3,222 (9.2%) | 9,498 (6.1%) | <0.001 |
| | No COVID infection | COVID-19 infection in 2020 | Total | p-value |
| | | Moderate symptoms | Severe symptoms | | |
| N | 119,424 (77.3%) | 19,062 (12.3%) | 16,079 (10.4%) | 154,565 (100.0%) | |
| Incidence of depression or/and anxiety in 2021 | 6,276 (5.3%) | 1,586 (8.3%) | 1,636 (10.2%) | 9,498 (6.1%) | <0.001 |

After controlling for potential confounders, subjects with COVID-19 infection had a 49% higher likelihood of receiving new depression and anxiety diagnoses in 2021 (OR = 1.49, p<0.01). Compared to the 6–10 age group, the 11–15 age group had 15% higher odds of new depression and anxiety (OR = 1.15, p<0.01). Subjects with sleep disorders (OR = 1.47, p<0.01), ADHD (OR = 1.51, p<0.01), pain diagnoses (OR = 1.23, p<0.01), and headaches (OR = 1.25, p<0.01) had higher odds of depression and anxiety in 2021 (Table 3).

**Table 3. Factors associated with the incidence of depression and anxiety.**

| Variable | Odds ratio | p-value | 95% confidence interval | |
|---|---|---|---|---|
| COVID-19 infection | 1.49 | <0.01 | 1.42 | 1.57 |
| Age category | | | | |
| 6-10 | reference | | | |
| 11-15 | 1.15 | <0.01 | 1.14 | 1.16 |
| Female | 1.91 | <0.01 | 1.82 | 2.01 |
| Type of insurance | | | | |
| Private insurance | reference | | | |
| Medicaid | 0.83 | <0.01 | 0.77 | 0.89 |
| Race/Ethnicity | | | | |
| Non-Hispanic White | reference | | | |
| Hispanic | 0.66 | <0.01 | 0.59 | 0.73 |
| Non-Hispanic Black | 0.55 | <0.01 | 0.42 | 0.73 |
| Non-Hispanic others | 0.35 | <0.01 | 0.27 | 0.45 |
| Unknown | 0.70 | <0.01 | 0.65 | 0.75 |
| Sleep disorder | 1.47 | <0.01 | 1.26 | 1.72 |
| ADHD | 1.51 | <0.01 | 1.27 | 1.78 |
| Pain | 1.23 | <0.01 | 1.16 | 1.30 |
| Blindness | 1.06 | 0.35 | 0.90 | 1.24 |
| Cancer | 1.14 | <0.01 | 1.04 | 1.25 |
| Asthma | 1.16 | 0.01 | 1.04 | 1.29 |
| Headache | 1.25 | <0.01 | 1.11 | 1.40 |
| Severe allergy | 1.13 | 0.01 | 1.04 | 1.23 |
| Hearing loss | 1.06 | 0.63 | 0.83 | 1.36 |
| Neurocognitive disorders | 1.34 | <0.01 | 1.16 | 1.56 |
| Obesity | 1.37 | <0.01 | 1.24 | 1.50 |

Compared to those without COVID-19 infection, subjects with moderate symptoms and those with severe symptoms were 40% (OR = 1.40, p < 0.01) and 59% (OR = 1.59, p < 0.01) more likely to experience depression and anxiety, respectively. Females had a 91% higher likelihood of having depression and anxiety than males (OR = 1.91, p < 0.01), and those with sleep disorders had 47% higher odds of experiencing these conditions (OR = 1.47, p < 0.01) (Table 4).

mental illness of youth as most studies focus on the impacts of the COVID-19 infection among adult populations (≥18 years old), [20,21]. In addition, studies have explored the broader effects of the COVID-19 pandemic on children, including such variables as school closures, reduced social interactions, changes in family dynamics, mental health, and financial instability [3,4,7,22–27].

Fewer studies have examined the direct impact of COVID-19 infection on individuals, especially on children and youth. Some studies have found no significant impact of COVID-19 on children's mental health, while other studies reported that COVID-19 infection may lead to a higher likelihood of youth being diagnosed with a mental disorder. Using electronic medical records of youth from 12–17 years old in 2020 and 2021, Bilu et al. (2023) reported that infected youth had significantly lower risks in depression (hazards ratio (HR)=0.65, 95% CI: 0.53–0.80, p < 0.01) and anxiety (HR = 0.82, 95% CI: 0.71–0.95, p < 0.01) than youth with no COVID-19 infection. Taquet et al. (2022) using a retrospective cohort study design, reported that children with and without COVID-19 infection had no difference in incidence of anxiety disorder (HR = 1.00, 95% CI: 0.94–1.06) and mood disorder (HR = 1.02, 95% CI: 0.94–1.10) [28]. On the other hand, some studies have assessed the psychological impact of COVID-19 on children and adolescents, highlighting the increased risk of

**Table 4. Association between COVID-19 infection severity and the incidence of depression and anxiety.**

| Variable | Odds ratio | p-value | 95% confidence interval | |
|---|---|---|---|---|
| Severity of COVID-19 infection | | | | |
| No infection | reference | | | |
| Moderate infection | 1.40 | <0.01 | 1.32 | 1.49 |
| Severe infection | 1.59 | <0.01 | 1.50 | 1.70 |
| Age category | | | | |
| 6-10 | reference | | | |
| 11-15 | 1.15 | <0.01 | 1.14 | 1.16 |
| Female | 1.91 | <0.01 | 1.82 | 2.01 |
| Type of insurance | | | | |
| Private insurance | reference | | | |
| Medicaid | 0.83 | <0.01 | 0.77 | 0.89 |
| Race/Ethnicity | | | | |
| Non-Hispanic White | reference | | | |
| Hispanic | 0.66 | <0.01 | 0.59 | 0.73 |
| Non-Hispanic Black | 0.55 | <0.01 | 0.42 | 0.73 |
| Non-Hispanic others | 0.35 | <0.01 | 0.27 | 0.45 |
| Unknown | 0.70 | <0.01 | 0.65 | 0.76 |
| Sleep disorder | 1.47 | <0.01 | 1.26 | 1.72 |
| ADHD | 1.51 | <0.01 | 1.28 | 1.79 |
| Pain | 1.22 | <0.01 | 1.15 | 1.30 |
| Blindness | 1.06 | 0.51 | 0.90 | 1.24 |
| Cancer | 1.14 | 0.01 | 1.04 | 1.25 |
| Asthma | 1.16 | 0.01 | 1.04 | 1.29 |
| Headache | 1.24 | <0.01 | 1.10 | 1.40 |
| Severe allergy | 1.13 | 0.01 | 1.04 | 1.23 |
| Hearing loss | 1.05 | 0.67 | 0.83 | 1.35 |
| Neurocognitive disorders | 1.34 | <0.01 | 1.15 | 1.55 |
| Obesity | 1.36 | <0.01 | 1.24 | 1.50 |

mental health disorders, including anxiety and depression, following infection. A study by McVoy et al. (2023) examined pediatric subjects with chronic illnesses, such as asthma and diabetes, and found that these children were more likely to develop a new psychiatric disorder following a COVID-19 infection. Specifically, the study found a positive association between asthma and the development of a new psychiatric diagnosis, with an odds ratio (OR) of 1.4 (95% CI: 1.15–1.71, p < .001). Children with diabetes exhibited an even stronger association, with an OR of 1.81 (95% CI: 1.11–2.94, p = .015) [5]. Similarly, Liu et al. (2021) explored the mental health outcomes of children who had recovered from COVID-19, finding that 31.6% of the participants reported significant anxiety symptoms, almost double the rate observed in healthy children (18.9%). Among the 38 pediatric patients they studied, 18.4% exhibited clinically significant PTSD symptoms, and 15.8% showed signs of depression. The study also highlighted a significant association between the severity of COVID-19 symptoms and the level of anxiety and depression, with children experiencing more severe symptoms tending to report higher levels of mental distress [8].

Furthermore, Solmaz et al. (2022) investigated anxiety levels in 292 pediatric COVID-19 patients (8–18 years old in 2020 and 2021) and found that 49.3% of the children reported high anxiety levels, which were primarily attributed to social isolation and the fear of death or family loss. The mean anxiety score for the participants was 25.5 (SD = 14.37), with girls

(mean = 29.54) showing significantly higher anxiety levels compared to boys (mean = 21.71). The study also observed a direct correlation between the number of symptoms and anxiety scores, with children experiencing 11 or more symptoms reporting the highest anxiety levels (mean = 30.89), compared to those with 0–5 symptoms (mean = 22.42) [29]. Strawn et al. (2023) examined differences in anxiety symptoms of adolescents (12–17-year-old in 2019) before and after COVID-19 infection. They reported the increase in anxiety symptomology to be 33% in through the years following COVID-19 infection.

Serious physical illnesses often correlate with adverse psychological outcomes [30–32]. At the onset of the pandemic, healthcare efforts primarily focused on treating the physical health symptoms from COVID-19 infection. The novelty of the virus meant that the psychological impact of the illness took a lower priority compared to the discovery of its cure, leaving a gap in mental health care. However, research has now highlighted the needs for also treating the mental health symptoms with a focus on vulnerable populations. Health crises can disproportionately impact younger populations, exacerbating mental health vulnerabilities through the stress of physical illness [33–37]. The direct experience of illness, fear of severe outcomes, the stigma associated with being infected [11,38,39], fatality of a loved one [40], may exacerbate the distress caused by the infection. Youth who were directed to emergency room services and hospitalizations for severe infections likely faced even more acute distress, amplifying their risk of mental health disorders [30–32]. Some individuals with COVID-19 infections may experience long periods of illness, which can persist for more than three months and pose additional challenges in managing their mental health. Additionally, fatigue, sleep disturbances, depression, anxiety, psychosis, cognitive impairment, obsessive-compulsive disorder, and post-traumatic stress disorder were some noted mental health conditions in individuals following long COVID [41]. All these contexts may act as stressors that are particularly challenging for children and adolescents, as they are in a crucial stage of development [34,35,42].

This study found that approximately 46% of youth with COVID-19 experienced severe infection, defined as hospital admissions or ER visits within seven days before or after infection. However, other studies, which used varying definitions of COVID-19 severity, reported lower rates. For instance, Mayo Clinic (2024) found a prevalence of 1.5% of severe cases among youth [43]. Similarly, another study reported a prevalence of 5.2%, with the majority of cases being mild or moderate [44]. In contrast, Campos et al. (2024) found a prevalence of 28%, with significant variations in disease severity based on comorbidities and other factors [45]. Unlike these prior studies, the findings suggest that the proportion of youth with severe COVID-19 infection may be higher than previously reported. This discrepancy highlights the potential for an underestimation of mental illness among children and adolescents affected by the pandemic. For example, in Campos et al. (2024), the presence of dyspnea was significantly associated with more severe disease, with an odds ratio (OR) of 6.3 (95% CI: 2.8–14.3), while fever (OR=3.8; 95% CI: 2.0–7.4) and cough (OR=3.4; 95% CI: 2.0–6.0) were similarly linked to greater disease severity [45]. These findings underscore the need for further investigation into the mental health impacts on children and adolescents, particularly those with severe COVID-19 infection or comorbid conditions. Such insights are crucial for developing comprehensive care strategies for this vulnerable population.

The purpose of examining a population-level data such as Utah APCD was to highlight more nuanced impacts to children's mental health from COVID-19 infection and characterize the need for mental health support and a broader public health response to care for vulnerable populations [46]. This study exhibits the bio-psycho-social impact of COVID-19 and should serve to inform the continued treatment of COVID-19 as a holistic process. Healthcare providers, educators, and policymakers must recognize that the psychological impact of COVID-19 infections extends beyond the acute phase of infection, particularly for those who experienced severe illness. Increased funding for mental health services is essential to meet the growing demand for support. Mental health interventions tailored for youth, especially those recovering from severe COVID-19 infections, are essential to mitigate the long-term psychological consequences. Early identification of mental health needs, particularly in children with severe infections, is essential to prevent long-term impairments and promote recovery. Sufficient and equitable care should be allotted to all parties suffering from the COVID-19 pandemic.

Additional consequences brought on by COVID-19 infection, such as missing school days, reduced social interactions, and the challenge of catching up academically upon returning to school, likely compound the psychological impact of COVID-19

infection [3,4,22,47,48]. Schools and community organizations must play a pivotal role in supporting students with mental health challenges considering this current body of research [43,49,50]. For instance, developing inclusive programs that offer psychological counseling, peer support, academic interventions, and mental health education may help foster resilience and academic success in youth affected by the COVID-19 infection. Additionally, those providing mental health interventions should account for the unique aspects of the population they serve. Factors such as the severity of COVID-19 infection, the age of youths, and the disproportionate impact of mental illness on racial and ethnic minorities must be considered [51–55].

This research fills a specific gap within the literature which is invaluable to informing medical providers and school personnel. However, several limitations must be acknowledged. First, the study relies on claims data, which may not capture subclinical symptoms of depression and anxiety that do not result in formal diagnoses. It is unclear from claims data whether the new diagnoses of depression or anxiety were situational, i.e., reactive to the stresses of illness or the pandemic environment, endogenous – emerging from biological or psychological predisposition, or related to medical factors (e.g., post-infection inflammatory effects or medication side effects). Our administrative data lack the clinical detail to attribute cause. Therefore, while we observe an association between COVID-19 infection and subsequent mental health diagnoses, we cannot discern the underlying pathway, whether these conditions arose directly from biological effects of the virus, indirectly from psychosocial stressors, or from the experience of illness and treatment. Second, the reliance on ICD-10 codes for identifying mental health conditions could introduce some degree of misclassification. Another important limitation is that medications were not used to identify or verify comorbid conditions, which may have resulted in underreporting or incomplete identification of health conditions including mental disorders. In addition, many children and adolescents may experience no symptoms despite having COVID-19 infections, resulting in the absence of COVID-19-related diagnosis and procedure codes. Consequently, there may be an underestimation of the number of individuals with COVID-19 infections. The generalizability of findings may be limited to Utah, as healthcare policies and demographics vary by state. Additionally, our analysis excluded uninsured children, potentially underestimating the true incidence of depression and anxiety. Future studies should include diverse populations and uninsured youth to strengthen generalizability. Furthermore, there were considerable amounts of missing data on race and ethnicity, which limits the ability to explore mental health outcomes across these important demographic factors. While efforts were made to account for various confounders through covariates, the study cannot entirely rule out the influence of unmeasured factors such as family history of mental illness, parental stress, or community-level stressors. Additionally, our claims data do not capture important contextual factors such as the severity of psychiatric symptoms, exposure to psychosocial stressors, or family history of mental illness. Because the APCD is designed for administrative tracking of healthcare utilization, some clinical nuances are not available. This means our findings must be interpreted as population-level associations, with caution regarding individual-level clinical inference. In addition, this study used administrative claims diagnoses to identify depression and anxiety, which means that a recorded diagnosis might sometimes reflect a provisional assessment or a billing code rather than a formally confirmed psychiatric evaluation. There is potential for misclassification; for example, a code could be used to justify a referral or to rule out a condition without indicating a definitive diagnosis. As a result, some children labeled with 'depression' or 'anxiety' in claims data might not meet full clinical diagnostic criteria, while conversely, some with true symptoms might not be coded if they did not seek care. Lastly, as the data only extends through 2021, the long-term psychological impact of COVID-19 on children and adolescents remains uncertain.

## Conclusions

This study provides evidence that COVID-19 infection, particularly severe cases, is associated with a higher incidence of depression and anxiety among children and adolescents. These findings highlight the need for targeted mental health interventions for youth affected by the pandemic. As society continues to navigate the post-pandemic landscape, prioritizing the mental well-being of younger populations is critical for fostering resilience and ensuring that adequate resources are available to support their psychological recovery. Addressing these challenges now will not only benefit individual children and adolescents but will also contribute to the broader goal of promoting public mental health in future generations.

## Informed consent statement

Patient consent was waived for this study because it poses minimal risk to participants and aligns with ethical guidelines for research.

## Supporting information

**S1 Appendix. Standardized mean differences before and after entropy balancing.**
(DOCX)

## Author contributions

**Conceptualization:** Jaewhan Kim, Aaron Fischer.

**Data curation:** Jaewhan Kim.

**Formal analysis:** Jaewhan Kim.

**Funding acquisition:** Fernando Wilson.

**Investigation:** Jaewhan Kim, Carson R. Ewing, Ashlee Larson, Chathuri Illapperuma-Wood, Aaron Fischer, Emeka Elvis Duru, Youngwoo Kim, Fernando Wilson.

**Methodology:** Jaewhan Kim.

**Project administration:** Chathuri Illapperuma-Wood, Emeka Elvis Duru, Fernando Wilson.

**Validation:** Jaewhan Kim.

**Writing – original draft:** Jaewhan Kim, Carson R. Ewing, Ashlee Larson, Chathuri Illapperuma-Wood, Aaron Fischer, Emeka Elvis Duru, Youngwoo Kim, Fernando Wilson.

**Writing – review & editing:** Jaewhan Kim, Carson R. Ewing, Ashlee Larson, Chathuri Illapperuma-Wood, Aaron Fischer, Emeka Elvis Duru, Youngwoo Kim, Fernando Wilson.

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
