## [Decision Letter · Decision Letter 0]

23 Jul 2025

Dear Dr. Kim,

Thank you for submitting your manuscript to PLOS ONE. After careful consideration, we feel that it has merit but does not fully meet PLOS ONE’s publication criteria as it currently stands. Therefore, we invite you to submit a revised version of the manuscript that addresses the points raised during the review process.

**ACADEMIC EDITOR:**
**Please address the reviewer comments and resubmit for further consideration for publication**

We look forward to receiving your revised manuscript.

Kind regards,

Souparno Mitra, M.D.

Academic Editor

PLOS ONE

Additional Editor Comments (if provided):

Reviewers' comments:

Reviewer's Responses to Questions

**Comments to the Author**

1. Is the manuscript technically sound, and do the data support the conclusions?

Reviewer #1: Yes

Reviewer #2: Yes

2. Has the statistical analysis been performed appropriately and rigorously?

Reviewer #1: Yes

Reviewer #2: I Don't Know

3. Have the authors made all data underlying the findings in their manuscript fully available?

Reviewer #1: Yes

Reviewer #2: Yes

4. Is the manuscript presented in an intelligible fashion and written in standard English?

Reviewer #1: Yes

Reviewer #2: Yes

Reviewer #1: Subjects with Medicare(disability) were excluded?

It is worth noting that ICD-10 code F34.1 (dysthymia) requires a minimum symptom duration of two years, which corresponds to the entire study period. This raises questions about the timing and onset of the condition relative to the study window

The use of ICD-10 codes to define the primary dependent variables (depression and anxiety) is noted; however, the absence of detail on the clinical criteria or assessment methods used to assign these codes introduces some uncertainty regarding diagnostic validity.

F10 Alcohol related disorders may not be relevant in this population.

Clarifying whether symptoms reflect situational, endogenous, or medically induced depression or anxiety (e.g., due to COVID-19) would significantly enhance the depth of the findings.

Reviewer #2: Some comments-

- The APCD is not designed for clinical research and lacks information on symptom severity, duration, functional impairment. It also wont analyze Psychosocial stressors (e.g., school closures, family dynamics, loss, trauma) or Family psychiatric history.

- Although 70 % is a high percent- it excluded the uninsured children which still make a chunk of the data. As a result the data set may under estimate the true incidence of depression/anxiety.

-Mental health diagnoses in children may be under reported due to stigma, or lack of access to specialty care. Diagnoses from Databases may be used solely for billing purposes and not reflect a confirmed clinical diagnosis.

- Any reason that Adolescent age group wasn't extended to 19 yrs of age rather than just 17 yrs ? The period upto 19 would have been significant due to added stressors of life changes such as moving out of the parental home, college etc

**Do you want your identity to be public for this peer review?** For information about this choice, including consent withdrawal, please see our Privacy Policy

Reviewer #1: **Yes: ** Anoop Narahari

Reviewer #2: **Yes: ** Arun Prasad

---

## [Author Response · Author response to Decision Letter 1]

9 Aug 2025

August 06, 2025

Dear Dr. Mitra and Reviewers,

We appreciate the opportunity to revise our submitted manuscript, Incidence of depression and anxiety in children and adolescents following COVID-19 infection. The comments are very helpful to improve the manuscript. Below are the specific reviewer’s comments (in bold) followed by our responses. Newly added parts in the manuscript are highlighted in yellow. Thank you again for taking the time to consider our manuscript.

Reviewers' comments:

Reviewer #1:

1. Subjects with Medicare(disability) were excluded?

Response: We appreciate the reviewer’s attention to this detail. We focused on children and adolescents aged 6 to 15 years in 2019, who were continuously enrolled in private insurance or Medicaid. The study population did not include subjects with Medicare (i.e. dual eligible), as the pediatric population with Medicare is extremely small (usually <0.5%) and generally represents unique clinical situations not reflective of the general child and adolescent population. We clarified this in the Subjects section:

“Children and adolescents who were eligible for Medicare due to disability (i.e. dual eligible) were excluded from this analysis, as the pediatric Medicare population is extremely small and typically consists of individuals with rare and complex clinical conditions that are not representative of the general child and adolescent population.”

2. It is worth noting that ICD-10 code F34.1 (dysthymia) requires a minimum symptom duration of two years, which corresponds to the entire study period. This raises questions about the timing and onset of the condition relative to the study window

Response: Thank you for the comment. We agree with the reviewer that ICD-10 code F34.1 (dysthymic disorder) denotes a chronic depressive condition typically requiring a prolonged duration of symptoms (approximately two years, or one year in children/adolescents). We have removed F34.1 (dysthymia) from the list of outcome codes used to define new-onset depression. This ensures that our outcome definition focuses on acute or subacute depressive disorders that can plausibly develop within our observation window.

In the methods section where we describe mental health outcome definitions, we revised the text to read:

“The primary dependent variable assessed was the occurrence of depression and anxiety in 2021, assessed as a binary variable (Yes/No). Depression and anxiety diagnoses were identified using ICD-10 codes according to the Centers for Medicare and Medicaid Services (CMS) and Agency for Healthcare Research and Quality (AHRQ) Clinical Classifications Software Refined (CCSR) guidelines [15-16]. Depression was identified using ICD-10 codes (F06.31-F06.34, F32-F33.9), explicitly excluding F34.1 (dysthymia) due to its diagnostic requirement of prolonged symptom duration which exceeds the one-year follow-up of our study. Anxiety was identified using ICD-10 codes (F06.4, F12.180-F16.980, F18.180-F19.980, F40.00-F41.9, F93.0, F94.0). Alcohol-related disorders (ICD-10 codes F10.180-F10.980) were excluded due to their limited clinical relevance within the studied pediatric population."

In 2021, 14 children and adolescents were diagnosed with dysthymia (ICD-10 code F34.1), with 6 in the COVID-19 infection group and 8 in the non-infection group. These individuals were excluded from the analysis, and the corresponding numbers in the tables were updated accordingly. These revisions have also been reflected in the Results section. However, the exclusion did not lead to any significant changes in the findings, including the odds ratios for the controlled variables.

3. The use of ICD-10 codes to define the primary dependent variables (depression and anxiety) is noted; however, the absence of detail on the clinical criteria or assessment methods used to assign these codes introduces some uncertainty regarding diagnostic validity.

Response: We appreciate this request for clarification. In our approach, we relied on established classification systems to define depression and anxiety diagnoses from ICD-10 codes. We used code groupings aligned with the Clinical Classification Software Refined (CCSR) developed by AHRQ and CMS for mental health conditions, which provides a clinically vetted mapping of ICD-10 codes into diagnostic categories.

4. F10 Alcohol related disorders may not be relevant in this population.

Response: Thank you for the comment. We agree with the reviewer that alcohol-related disorder codes (ICD-10 F10.x) have very limited applicability in a pediatric population. In our initial analysis, F10 codes were included as part of a broader list of mental health comorbidities; however, we found that virtually no participants under 18 had an F10 diagnosis in the dataset.

When we examined the raw data—including individuals without continuous insurance coverage—we identified 52 adolescents with alcohol-related disorder diagnoses. These individuals, however, were excluded from the final analysis because they did not meet our inclusion criteria, which required continuous insurance coverage for three years (2019–2021).

To avoid confusion, we have removed mention of F10 alcohol-related disorders from the manuscript. Revision in the outcome session has been made, alongside with comment 2.

“Alcohol-related disorders (ICD-10 codes F10.xx) were excluded due to their limited clinical relevance within the studied pediatric population.”

5. Clarifying whether symptoms reflect situational, endogenous, or medically induced depression or anxiety (e.g., due to COVID-19) would significantly enhance the depth of the findings.

Response: Thank you for the comment. We acknowledge the importance of distinguishing the context of post-COVID depression and anxiety, though our data are limited in this regard. Because we used ICD-10 diagnosis codes to define depression and anxiety after COVID-19 infection, these codes do not reliably indicate whether symptoms are situational, endogenous, or medically induced by the COVID-19 infection itself. Given our study design, we can only report on the observed association between COVID-19 infection and the incidence of depression and/or anxiety, without identifying the underlying etiology.

We have added language in the Discussion (limitations section) to clarify that our study cannot definitively determine the origin or context of the diagnosed mental health conditions.

We now state: “It is unclear from claims data whether the new diagnoses of depression or anxiety were situational, i.e. reactive to the stresses of illness or the pandemic environment, endogenous - emerging from biological or psychological predisposition, or related to medical factors (e.g., post-infection inflammatory effects or medication side effects). Our administrative data lack the clinical detail to attribute cause. Therefore, while we observe an association between COVID-19 infection and subsequent mental health diagnoses, we cannot discern the underlying pathway, whether these conditions arose directly from biological effects of the virus, indirectly from psychosocial stressors, or from the experience of illness and treatment.”

Reviewer #2: Some comments-

1. The APCD is not designed for clinical research and lacks information on symptom severity, duration, functional impairment. It also won’t analyze Psychosocial stressors (e.g., school closures, family dynamics, loss, trauma) or Family psychiatric history.

Response: Thank you for the comment. We fully agree with this comment by the reviewer. This is a limitation of using administrative claims data and is now emphasized in the Limitations section.

… “Additionally, our claims data do not capture important contextual factors such as the severity of psychiatric symptoms, exposure to psychosocial stressors, or family history of mental illness. Because the APCD is designed for administrative tracking of healthcare utilization, some clinical nuances are not available. This means our findings must be interpreted as population-level associations, with caution regarding individual-level clinical inference.”

2. Although 70 % is a high percent- it excluded the uninsured children which still make a chunk of the data. As a result the data set may underestimate the true incidence of depression/anxiety.

Response: Thank you for the comment. The reviewer is correct that our study includes only children and adolescents with continuous health insurance coverage; therefore, uninsured children are not represented, which is an inherent limitation of using claims data. This could indeed lead to an underestimation of the true incidence of post-COVID depression and anxiety in the general population if uninsured children have different risk profiles or less access to care.

We have added a statement to the Limitations section highlighting this issue:

“The generalizability of findings may be limited to Utah, as healthcare policies and demographics vary by state. Additionally, our analysis excluded uninsured children, potentially underestimating the true incidence of depression and anxiety. Future studies should include diverse populations and uninsured youth to strengthen generalizability”

3. Mental health diagnoses in children may be under reported due to stigma, or lack of access to specialty care. Diagnoses from Databases may be used solely for billing purposes and not reflect a confirmed clinical diagnosis.

Response: We appreciate this important point. Diagnoses in claims data are indeed recorded primarily for billing and administrative purposes and may not always undergo the same confirmation as diagnoses established in structured research settings or through standardized clinical interviews.

We have strengthened our discussion of this limitation. The following statement has now been added to the Discussion section:

“In addition, this study used administrative claims diagnoses to identify depression and anxiety, which means that a recorded diagnosis might sometimes reflect a provisional assessment or a billing code rather than a formally confirmed psychiatric evaluation. There is potential for misclassification; for example, a code could be used to justify a referral or to rule out a condition without indicating a definitive diagnosis. As a result, some children labeled with ‘depression’ or ‘anxiety’ in claims data might not meet full clinical diagnostic criteria, while conversely, some with true symptoms might not be coded if they did not seek care.”

4. Any reason that Adolescent age group wasn't extended to 19 yrs of age rather than just 17 yrs? The period upto 19 would have been significant due to added stressors of life changes such as moving out of the parental home, college etc

Response: We thank the reviewer for this suggestion. The study population consisted of school-aged children and adolescents in Utah, from elementary to high school age, who were 6 to 15 years old in 2019 and had continuous health insurance enrollment from 2019 through 2021. Participants who were 15 years old in 2019 reached age 17 (high school) by 2021, thus remaining within the broader adolescent school-age category.

Response: The manuscript adheres to all PLOS ONE style requirements.

Response: Due to legal and ethical restrictions imposed by the Utah Department of Health and Human Services (third-party organization), the University of Utah’s Institutional Review Board, and the Utah Resource for Genetic and Epidemiologic Research (RGE), we are unable to publicly share even de-identified individual-level data used in this study. Specifically, these data contain potentially identifying and sensitive patient health information protected under state regulations and institutional policies. Data access is managed by the RGE, and requests for data can be directed to the University of Utah Institutional Review Board/Data Access Committee via the RGE website (https://rge.utah.edu) or through direct contact at:

Institutional Review Board (IRB) Office

University of Utah

75 South 2000 East, Salt Lake City, Utah 84112

Phone: 801-581-3655

Email: irb@hsc.utah.edu

Utah Resource for Genetic and Epidemiologic Research (RGE)

University of Utah

Research Administration Building, 75 S 2000 E room 207, Salt Lake City, UT 84132

Phone: 801-581-6351

Email: rge@hsc.utah.edu

Response: We have moved the ethics statement to the Methods section and removed it from all other sections of the manuscript, in accordance with journal guidelines.

Response: No specific previously published works were recommended for citation by the reviewers.

Response: We have reviewed the reference list and confirm that it is complete, accurate, and does not include any retracted publications.

---

## [Decision Letter · Decision Letter 1]

24 Aug 2025

Incidence of depression and anxiety in children and adolescents following COVID-19 infection

PONE-D-25-32255R1

Dear Dr. Kim,

We’re pleased to inform you that your manuscript has been judged scientifically suitable for publication and will be formally accepted for publication once it meets all outstanding technical requirements.

Kind regards,

Souparno Mitra, M.D.

Academic Editor

PLOS ONE

Additional Editor Comments (optional):

Reviewers' comments:

Reviewer's Responses to Questions

**Comments to the Author**

Reviewer #1: All comments have been addressed

Reviewer #2: All comments have been addressed

2. Is the manuscript technically sound, and do the data support the conclusions?

Reviewer #1: Yes

Reviewer #2: Yes

3. Has the statistical analysis been performed appropriately and rigorously?

Reviewer #1: Yes

Reviewer #2: I Don't Know

4. Have the authors made all data underlying the findings in their manuscript fully available?

Reviewer #1: Yes

Reviewer #2: Yes

5. Is the manuscript presented in an intelligible fashion and written in standard English?

Reviewer #1: Yes

Reviewer #2: Yes

Reviewer #1: All of my comments were addressed by authors. These revisions have also been reflected in the

Results section

Reviewer #2: Thank you for the edits and responses towards suggestions for the paper "Incidence of depression and anxiety in children and adolescents following COVID-19 infection."

**Do you want your identity to be public for this peer review?** For information about this choice, including consent withdrawal, please see our Privacy Policy

Reviewer #1: **Yes: ** Anoop Narahari

Reviewer #2: **Yes: ** Arun George Prasad

---

## [Editor Report · Acceptance letter]

PONE-D-25-32255R1

PLOS ONE

Dear Dr. Kim,

I'm pleased to inform you that your manuscript has been deemed suitable for publication in PLOS ONE. Congratulations! Your manuscript is now being handed over to our production team.

Kind regards,

on behalf of

Dr. Souparno Mitra

Academic Editor

PLOS ONE